# Opinion: A research roadmap for exploring atmospheric methane removal via iron salt aerosols

Katrine A. Gorham[1], Sam Abernethy[1], Tyler R. Jones[2], Peter Hess[3], Natalie M. Mahowald[4], Daphne Meidan[5], Matthew S. Johnson[6], Maarten M.J.W. van Herpen[7], Yangyang Xu[8], Alfonso Saiz-Lopez[5], Thomas Röckmann[9], Chloe A. Brashear[9], Erika Reinhardt[1], David Mann[1]

[1]Spark Climate Solutions, Covina, CA 91723, USA
[2]Institute of Arctic and Alpine Research, University of Colorado, Boulder, CO 80303, USA
[3]Department of Biological and Environmental Engineering, Cornell University, Ithaca, NY 14853, USA
[4]Department of Earth and Atmospheric Sciences, Cornell University, Ithaca, NY 14853, USA
[5]Department of Atmospheric Chemistry and Climate, Institute of Physical Chemistry Blas Cabrera, Spanish National Research Council, 28006 Madrid, Spain
[6]Department of Chemistry, University of Copenhagen, DK-2100 Copenhagen Ø, Denmark
[7]Acacia Impact Innovation, Bernheze 5384 BB, The Netherlands
[8]Department of Atmospheric Sciences, Texas A&M University, College Station, TX, 77843, USA
[9]Institute for Marine and Atmospheric Research Utrecht, Department of Physics, Faculty of Science, Utrecht University, 3584 CS Utrecht, The Netherlands

*Correspondence to*: Katrine A. Gorham (katrine@sparkclimate.org)

**Abstract.** The escalating climate crisis requires rapid action to reduce the concentrations of atmospheric greenhouse gases and lower global surface temperatures. Methane will play a critical role in near-term warming due to its high radiative forcing and short atmospheric lifetime. Methane emissions have accelerated in recent years and there is significant risk and uncertainty associated with the future growth in natural emissions. The largest natural sink of methane occurs through oxidation reactions with atmospheric hydroxyl and chlorine radicals. Enhanced atmospheric oxidation could be a potential approach to remove atmospheric methane. One method proposes the addition of iron salt aerosols (ISA) to the atmosphere, mimicking a natural process proposed to occur when mineral dust mixes with chloride from sea spray to form iron chlorides, which are photolyzed by sunlight to produce chlorine radicals. Under the right conditions, lofting ISA into the atmosphere could potentially reduce atmospheric methane concentrations and lower global surface temperatures. Recognizing that potential atmospheric methane removal must only be considered as an additive measure – in addition to, not replacing, crucial anthropogenic greenhouse gas emission reductions and carbon dioxide removal – roadmaps can be a valuable tool to organize and streamline interdisciplinary and multifaceted research to efficiently move towards understanding whether an approach may be viable and socially acceptable, or if it is nonviable and further research should be deprioritized. Here we present a       five-year research roadmap to explore whether ISA enhancement of the chlorine radical sink could be a viable and socially acceptable atmospheric methane removal approach.

## 1 Introduction

Driven by anthropogenic greenhouse gas emissions, Earth's average surface temperature has increased by 1.1 °C since 1850 (Forster et al., 2021). As global temperature increases, irreversible changes in the Earth system will likely occur, such as ice sheet collapses in Greenland and Antarctica, coral reef die-off, abrupt permafrost thaw, mountain glacier loss, and Amazon rainforest dieback (Lenton et al., 2008; McKay et al., 2022). To mitigate both near-term and long-term warming we must undertake rapid, sustained action to establish a diverse portfolio of approaches to slow and then reverse the increase of atmospheric greenhouse gas concentrations, ideally reducing them to preindustrial levels. Emissions reductions must be prioritized. In addition, as near-term warming threatens to trigger climate tipping points, negative emissions approaches may be used to remove greenhouse gases already in the atmosphere and counter rising natural and uncontrollable emissions.

Methane ($CH_4$) emissions have    contributed roughly 0.5 °C to global warming    relative to preindustrial times, second only to carbon dioxide ($CO_2$    ) (IPCC, 2023). Methane lasts roughly a decade in the atmosphere (UNEP, 2021), with a Global Warming Potential 83 times that of carbon dioxide over 20 years and 30 times that of carbon dioxide over 100 years (Forster et al., 2021). The concentration of methane in the atmosphere is over 2.5 times pre-industrial levels, and the growth rate has accelerated since 2006, with record increases in 2020 and 2021 (NOAA, 2023). Methane emissions come from anthropogenic sources (fossil fuel use, agriculture, waste and wastewater, biomass burning, etc.) and natural sources (wetlands, oceans, freshwaters, termites, permafrost, etc.), both of which are increasing (Jackson et al., 2020; Nisbet et al., 2023, Saunois et al., 2020). As the planet continues to warm and precipitation patterns change, natural methane emissions are expected to increase from wetlands, as well as from permafrost due to abrupt thaw, thermokarst lake formation and expansion, and bacterial processes (Dean et al., 2018; Neumann et al., 2019; Paudel et al., 2016; Peng et al., 2022; Zhang et al., 2023).    On our current trajectory, natural emissions are estimated to increase by ~30-200 Tg CH4/yr by 2100 (Zhang et al., 2023; Kleinen et al., 2021). There is evidence that wetland emissions increases are already underway, with roughly half of the 2020 atmospheric methane increase attributed to wetlands (Qu et al., 2022; Peng et al., 2022; Nisbet et al., 2023; Zhang et al., 2023).

The natural sinks for atmospheric methane are oxidation by gas phase radicals (~95 %) and uptake into soil by methanotrophic bacteria and archaea (~5 %) (Saunois et al., 2020). The atmospheric lifetime of methane is thus mainly determined by the oxidative capacity of the atmosphere. Approximately 550 Tg methane per year is removed by reactions with tropospheric hydroxyl radicals (OH.); tropospheric chlorine radicals (Cl·) destroy ~11 Tg methane per year (Saunois et al., 2020). However, the tropospheric chlorine radical sink estimate is poorly constrained, ranging from 1 to 35 Tg methane per year (Saunois et al., 2020). Recent research suggests the tropospheric    chlorine radical sink could be underestimated (van Herpen et al., 2023).

Complex and non-linear atmospheric chemistry dictates the oxidation of methane, which results in the production of formaldehyde and carbon monoxide which is further oxidized to carbon dioxide. Hydroxyl radical oxidation of methane can

result in the formation or loss of ozone, depending on the nitrogen oxide (NOx) concentrations. Chlorine radicals react to remove both methane and ozone, the principal precursor of hydroxyl radical production (Lelieveld et al., 2002; Seinfeld and Pandis, 2016). Therefore, chlorine radicals may either increase or decrease the atmospheric lifetime of methane depending on the concentration of chlorine radicals and atmospheric conditions (Li et al., 2022, van Herpen et al., 2023, Li et al., 2023). Volatile organic compounds and NOx will change the methane response since they directly and indirectly affect the concentrations of hydroxyl and chlorine radicals. Furthermore, since reactions with carbon monoxide and methane are the primary sinks for hydroxyl radicals, the abundance of methane impacts the oxidative character of the atmosphere in a self feedback process (Lelieveld et al., 2002; Staniaszek et al., 2022; Holmes, 2018    ).

Enhanced atmospheric oxidative sinks could increase the rate of atmospheric methane removal and therefore reduce near-term warming. While not a replacement for much needed anthropogenic emission reductions, enhancing atmospheric oxidative sinks may be an important negative emission approach considering projected ongoing elevated natural methane emissions. One untested proposal involves iron salt aerosols (ISA). This potential approach involves lofting iron-based particles into the troposphere (e.g., from ships or towers) to catalytically produce chlorine radicals (Oeste, 2009; Oeste et al., 2017), mimicking a natural phenomenon proposed to occur when mineral dust combines with sea spray aerosols (van Herpen et al., 2023). Discussing natural analogues of this process and the current state of research, this paper presents a roadmap for research and development that is needed to understand whether ISA enhancement of the chlorine radical sink may be a feasible, scalable, and safe approach for atmospheric methane removal.

## 2 State of the research

Currently, research into ISA falls into three categories: laboratory experiments to quantify the details of the mechanism, observational analysis of the natural analogue of ISA, known as mineral dust sea spray aerosol (MDSA), and numerical modeling evaluating the potential impacts of ISA.

### 2.1 Laboratory experiments

In a series of papers, Wittmer et al. demonstrated the production of chlorine atoms from iron-doped salts and aerosols (Wittmer et al., 2015a; Wittmer et al., 2015b; Wittmer et al., 2017). Reproducing and expanding upon these laboratory studies is of the highest importance, since the mechanism of chlorine generation is poorly understood (van Herpen et al., 2023; Wittmer et al., 2015; Wittmer et al., 2017; Zhu et al., 1997). The ISA mechanism is catalytic in iron and may be catalytic in chlorine in natural environments (Wittmer et al., 2017). Understanding of catalytic efficiency requires study of dependencies on conditions such as aerosol size distribution and number density; humidity; rate of supply of acidity to the system; rate of coagulation of aerosols; effects of organic chelating agents on iron activity; changes in chlorine escape probability due to aqueous chemical conditions; changes in iron activity due to aerosol chemistry; and, understanding how the atmosphere behaves under high ClOx

conditions (Pennacchio et al., submitted). Furthermore, studies of real atmospheric conditions are needed to better understand
suppression of the ISA mechanism in the presence of sulfate and oxalate (Wittmer et al., 2015; Wittmer et al., 2017).
**2.2 The natural analogue of ISA: mineral dust sea spray aerosol**
The natural mineral dust sea spray aerosol (MDSA) mechanism is proposed to occur when iron from mineral dust mixes with
the chlorine in sea spray, forming iron chlorides which are photolyzed to produce chlorine radicals (van Herpen et al., 2023).
ISA could mimic this natural MDSA phenomenon by only aerosolizing what is currently believed to be the key component of
the mineral dust: the photoactive iron.

van Herpen et al. (2023) provided the first evidence that methane may be removed over the North Atlantic by chlorine radicals
produced by the MDSA mechanism.     North African semi-arid regions are the dominant source of iron-containing mineral
dust, with frequent transport over the North Atlantic (Prospero et al., 2021). Using air samples collected in Barbados during
North African dust events     (Mak et al., 2003), a model parameterized with the MDSA mechanism of chlorine radical
production produced results consistent with a previously unexplained 13C-depletion in the reaction product carbon monoxide
(CO) (van Herpen et al., 2023). Carbon monoxide produced from chlorine radical oxidation of methane is extremely depleted
in 13C which makes $\delta 13C(CO)$ a very sensitive indirect detection method of chlorine radicals (Röckmann et al., 1999). The
variability of $\delta 13C(CO)$ in atmospheric air is the main evidence that the MDSA process is occurring, as there is no other
mechanism proposed that can explain the carbon monoxide isotope signature (Mak et al., 2003).

As a proxy for methane oxidation by chlorine radicals, studies of the isotopic composition of carbon monoxide in the mid-
Atlantic boundary layer are underway to further confirm the MDSA mechanism. This includes a regular air sampling program
at four ground-based stations in the North Atlantic (located in Barbados, Canary Islands, Cape Verde, and Brazil), as well as
Atlantic transect sampling onboard commercial vessels. Time series at the ground-based stations provide high-resolution
observations at the longitudinal margins of trans-Atlantic dust transport, including the site where seasonal depletions in
$\delta 13C(CO)$ were previously observed (Mak et al., 2003). Air samples collected in north-south ship transects may allow spatial
correlation of the $\delta 13C(CO)$ excursions with North African dust plumes. Together, these samples provide an opportunity to
investigate the formation process of MDSA, as well as the seasonal and spatial influence of North African dust on tropospheric
chlorine radical oxidation.
**2.3 Modeling**
Early modeling studies indicate that atmospheric chlorine additions can increase or decrease methane concentrations,
depending on the concentration of chlorine that is present in the atmosphere (Horowitz et al., 2020; Li et al., 2022; Saiz-Lopez
et al., 2023). Chlorine radicals readily oxidize ozone, the main precursor of hydroxyl radicals, which results in less methane
oxidation via hydroxyls; at low atmospheric concentrations, chlorine radicals reduce ozone concentrations without having a

commensurate impact on methane. Even though chlorine radicals react 16x faster with methane than the reaction of hydroxyl radicals with methane (Atkinson, 2006), hydroxyl radicals dominate the methane oxidation sink because they are much more abundant than chlorine radicals. As more chlorine is emitted, ozone concentrations will be reduced so that proportionally more chlorine radicals react with methane. The increased methane destruction by chlorine radicals will eventually outcompete decreased destruction by hydroxyl. In an initial, highly simplified model scenario, Li et al. (2023) found that a reduction in methane concentration could be achieved if more than 90 Tg Cl/yr (three times the estimated present-day emission rate) was added evenly to the atmosphere over all ocean surfaces, and lowering the global methane burden by 2,000 Tg would require the emission of an additional 1,000 Tg Cl2 /yr. However, assuming a uniform increase of chlorine over all ocean surfaces may underestimate the potential effectiveness of local chlorine radical generation where efficiency may be condition-dependent (e.g., $NO_x$, CO, and chlorine concentrations, humidity, temperature, altitude, etc.) (Meidan et al., submitted). For example, considering the MDSA natural analogue of ISA, van Herpen et al (2023) found that high dust concentrations in the North Atlantic corresponded with net methane removal, while globally lower dust concentrations led to a net increase of methane.

Current models may not accurately capture the speed and efficiency of producing chlorine radicals via the ISA mechanism due to assumptions of the percentage of photoactive iron in the emitted iron, aerosol pH, aerosol mixing rates, and more. Another challenge with Earth system models is that they instantaneously dilute emissions to model grid dimensions which could lead to underestimates or overestimates of the effectiveness of the ISA mechanism, especially when considering iron emission additions from point sources like ships (e.g. Meidan et al., submitted). For example, the mixing of the iron and sea salt within the aerosol is modeled to occur instantaneously (Meidan et al., submitted; van Herpen et al., 2023); however, in reality it would likely take hours to days, leading the global model to overestimate the rate of chlorine radical production. Furthermore, the ISA mechanism is likely to occur faster in high NOx environments (Oeste et al., 2017) but could be less efficient in high sulfate environments (Bondy et al., 2017; Chen et al., 2020; Legrand et al., 2017; Pio et al., 1998), and both NOx and sulfate may be co-emitted with iron (e.g. from a ship plume). Thus, models that instantaneously dilute emissions across the grid dimensions may misrepresent the ISA mechanism. Overall, it is unclear whether current Earth system models overestimate or underestimate the efficiency of the oxidation mechanism. Additional detailed box modeling focusing on deployment sites and constrained by field observations are necessary to assess the effectiveness of the mechanism. However, local box models are less reliable over timescales where mixing between different air masses is relevant. This motivates the need for high-resolution regional and seasonal modeling over ocean basins and variable resolution configurations embedded in global models.

Considering the difficulty of simulating variable atmospheric chemical conditions (e.g., atmospheric chemical composition, solar radiation, wind mixing, etc.) across different geographic locations, it is important to develop an ensemble of models that enable uncertainty assessment. Such a model ensemble will allow a comprehensive exploration of different iron salt aerosol deployment scenarios (magnitude each year), aerosol particle sizes, and deployment location and timing.

## 3 Roadmap

### 3.1 Roadmap framework

Roadmaps are used in climate research to define knowledge gaps, needs, and associated outputs as they relate to interdependencies and timelines, particularly in instances that benefit from integrated, interdisciplinary research. Recent examples include geochemical carbon dioxide removal (Masano et al., 2022), ocean-based carbon dioxide removal (Ocean Visions, 2023), ice sheet contributions to sea level rise (Aschwanden et al., 2021), and solar radiation management (Wanser et al., 2022). . A coordinated, thorough, and science-based approach is needed to ensure that resources are used efficiently, stakeholders and interdisciplinary teams are engaged on appropriate timelines, and efforts are focused towards sequenced research questions and milestones.

### 3.2 Viability assessment

The viability of an atmospheric methane removal approach can be assessed by considering its potential for feasibility, scalability, and social license to operate. A feasible approach must be climate beneficial, safe, acceptable for its side effects, and cost-plausible. Determination of scalability will be approach-specific, acknowledging that the scale of increased natural emissions is anticipated to be tens of millions of tons of methane per year (Kleinen et al., 2021).

The key milestone questions below can help determine the viability of ISA and whether it should continue to be prioritized. The research that informs the key milestones questions should be pursued in parallel (Table 1).

1. Is enhancement of the chlorine radical oxidative sink of methane via the ISA mechanism effective and climate beneficial? At what scale?
2. What impacts could the ISA approach have on Earth systems and human systems, both positive and negative? Is there a cost-plausible ISA deployment method?
3. What is needed to advance a structure of ethical governance and social license for utilizing atmospheric intervention to reduce atmospheric methane concentrations?

### 3.2.1 Milestone question #1: Is the ISA mechanism effective and climate beneficial and scalable?

The complexity and nonlinearity of atmospheric chemistry and meteorology requires laboratory, field, and plume and global modeling studies of the efficiency of chlorine radical production, its dependence on atmospheric conditions and other gases, and the impact on methane removal.

An important assumption in previous studies is that only 1.8 % of iron is photoactive (van Herpen et al., 2023; Meidan et al., submitted). However, the amount of ISA that is photoactive may vary by emission source (e.g. ship emissions may have more photoactive iron relative to mineral dust; Ito, 2013; Rodriguez et la., 2021), geographical location, aerosol pH, the

presence of other chemical constituents (e.g. sulfate and NOx), and altitude (Mahowald et al., 2018). Furthermore, the rate of
chlorine production - and subsequent rate of methane oxidation - per mass of photoactive iron is estimated to result in the
removal of 45 methane molecules per iron atom per day (van Herpen et al., 2023), but has many uncertainties including the
time that iron remains in the atmosphere which may be impacted by large regional variability in deposition rates (Meidan et
al., submitted). The efficiency, cost, safety (e.g. air quality), and net radiative forcing of ISA will depend on the percentage of
iron that is photoactive, the rates of chlorine production and methane oxidation per mass of photoactive iron, and  the lifetime
of photoactive-iron based aerosol.

Current studies assume that the chlorine radicals released from the photochemical reaction with iron will react (e.g., with
methane) to form hydrochloric acid, which will then be reabsorbed back into the aerosol and thus recycled (van Herpen et al.,
2023; Oeste et al., 2017). It is unclear under which atmospheric conditions this cycle occurs, but if some chlorine radicals are
lost     then the cycling would be less efficient. One potential mechanism by which the cycling efficiency could be reduced is
if hydrochloric acid is produced and deposited into the ocean         (Wang et al., 2019). Therefore, further laboratory
measurements and detailed box models are needed to further study hydrochloric acid recycling efficiency by ISA.

Better understanding of how sulfur dioxide and NOx concentrations impact the ISA mechanism is also needed (Oeste et al.,
2017). Sulfur dioxide and NOx concentrations vary regionally and locally and their emissions may sometimes coincide with
those of iron.

Smaller ISA particles have greater surface area to mass ratios and stay longer in the atmosphere, likely increasing the efficiency
of the ISA mechanism. Furthermore, smaller particles tend to be more acidic (Pye et al., 2020), which is required for the
mechanism to be active (Wittmer et al., 2015; 2017). However, smaller aerosols may be transported further to coastal or inland
locations where these particles could contribute directly to negative human health effects or deposit on terrestrial or ice-covered
surfaces with unintended consequences. The aerosol size may also affect marine cloud cover, thereby influencing local
radiative forcing. In addition, there tends to be more sulfate in smaller aerosols, which may reduce the effectiveness of the ISA
mechanism (Bondy et al., 2017; Chen et al., 2020; Legrand et al., 2017; Pio et al., 1998).

Studying MDSA (the natural analogue of ISA) through field studies is essential to understand the effectiveness of this
mechanism under different atmospheric conditions and its geographical extent, thereby better constraining atmospheric and
Earth system models. Early MDSA field studies are underway to explore the seasonality and spatial extent of methane
oxidation by chlorine radicals that may occur in natural dust plumes through proxy measurements of $\delta13C(CO)$. Further studies
– both natural analogue and *in-situ* ISA enhancement studies – would benefit from alternative ISA detection and quantification
approaches, including direct chlorine measurements or additional proxy measurements to reinforce existing observations.

Ideally, models will be developed to include the entire MDSA mechanism, including implementation of the isotope effect in the chlorine radical reaction with methane, thus enabling direct comparison of model results to observations of $\delta13C(CO)$. At present, some models (EMAC; Gromov 2014) include complete carbon monoxide isotope representation, but not the MDSA mechanism, whereas other models (CAM-CHem; van Herpen et al., 2023) include an initial representation of the MDSA mechanism but do not incorporate the isotopic effects.

Isotopic signatures and dust from ice core paleo records may elucidate evidence of historical MDSA activity. Methane isotope measurements of air trapped in polar ice cores have been used to constrain the methane budget in the past (Bock et al. 2017; Sapart et al., 2012; Fischer et al., 2008; Ferretti et al., 2005), but possible variations in the chlorine-based methane sink have not been taken into account in these studies. Dust levels have undergone strong changes in the past (e.g. Fischer et al., 2007; Han et al., 2018; Yuan et al., 2020; Yue et al., 2023), and associated changes in MDSA may have affected the paleo records of $\delta13C(CH4)$.

If modeling, laboratory studies, and natural analogue field studies prove that a promising, safe mechanism exists to produce chlorine radicals at sufficient scale and under diverse atmospheric conditions, it may be appropriate to consider field studies with intentional enhancement of ISA, using a broad suite of atmospheric measurements to understand how the mechanism performs in the real atmosphere (see Section 3.2.3     ). Prior to performing any ISA enhancement field studies – even at a small and controlled scale – it is essential to engage and work collaboratively with potentially impacted communities, policy and science leaders, governmental bodies, NGOs, media, and other stakeholders to ensure that actions are conducted with social license and an appropriate governance framework with free, prior, and informed consent (FAO, 2016). For example, a study could involve controlled enhancement of dust or emitted aerosolized iron, or could investigate existing anthropogenic emissions of iron (e.g. from a ship plume, power plant, iron foundry, etc). The scale of the study should be suitable to accommodate a likely non-linear atmospheric response, where substantial increases in chlorine – thus iron emissions – may be needed before there is a decrease in methane.

**3.2.2 Milestone question #2: What are the potential Earth system and human systems impacts of ISA, and is it cost-plausible?**

Lifecycle analyses are necessary to assess the potential benefits, tradeoffs, risks, uncertainties, and costs of ISA. As part of this analysis, understanding the potential impacts of ISA enhancement on the Earth system and human system is imperative before considering large scale deployment. Human system impacts may include human health outcomes, as well as indirect human impacts from Earth system changes. For example, if ISA resulted in ocean acidification there could be marine life implications resulting in     economic, biodiversity,     and cultural     impacts for coastal communities. Furthermore, if chlorine drifts into urban areas it could stimulate ozone formation and cause negative human health impacts (Wang et al., 2020).


Earth system modeling must be conducted to understand the impact of atmospheric conditions, aerosol size, and release
locations, magnitudes, and timing, on ocean systems, terrestrial systems, the cryosphere, the stratosphere, and clouds and
precipitation. Results from atmospheric modeling (plume and global), laboratory studies, and initial field studies will inform
priorities and research directions for Earth system studies and could potentially inform the design of field studies to verify
model results.

The potential broader impacts of ISA deployment beyond removing methane are not well understood. These may include co-
benefits such as reductions in atmospheric black carbon (Li et al., 2021, Oeste et al., 2017) and ozone (Li et al., 2023; van
Herpen et al., 2023), both of which are climate warming agents. However, there are also potential negative impacts that must
be further explored including ocean acidification, stratospheric ozone loss, adverse chemical side reactions (such as COCl2
formation), and albedo changes from deposition on ice surfaces (Li et al., 2023). Iron aerosols absorb light and thus tend to
warm the planet, offsetting some of the lowered radiative forcing from oxidized methane (Matsui et al., 2018; Li et al., 2021;
Meidan et al., submitted). There are also potential side effects such as indirect radiative forcing due to marine cloud brightening
and carbon dioxide absorption by ocean iron fertilization (Emerson, 2019) that could be either favorable or unfavorable.

The potential effects of ISA on air quality and human health are also poorly understood. Enhanced chlorine could lead to
beneficial reductions in tropospheric ozone. However, the reduced hydroxyl radical production may increase lifetimes of
atmospheric trace species that may be     detrimental to human health. Moreover, iron aerosols themselves present a human
health risk, especially when small (O'Day et al., 2022). Therefore, more research is required to determine under which
conditions (including deployment location/timing, particle composition, and ISA size) the air quality impact is beneficial or
detrimental.

Engineering deployment modalities and implementation scenarios is one of the later steps in this roadmap, only to be pursued
if earlier dependencies are addressed and ISA proves effective and climate beneficial with acceptable side effects.
Nevertheless, to avoid delaying potential future deployment-readiness and to iteratively refine design in advance of any *in-situ*
field studies, development and engineering of a nozzle sprayer delivery system could begin in parallel with early research
activities.

As part of a lifecycle analysis, the cost of materials (e.g., iron), infrastructure, and other implementation resources must be
assessed. To be cost-plausible, the     cost     per ton of methane removed     must have a viable path to becoming     lower
than the social cost of methane, a monetary valuation that estimates the socioeconomic impact caused by an additional metric
ton of methane (Azar et al., 2023).

**3.2.3 Milestone question #3: What is needed to advance ethical governance and social license?**

It is imperative that governance and social impacts be considered in parallel and iteratively with    the development    of any atmospheric methane removal approach. Addressing the climate crisis requires engagement beyond technical solutions; collaboration and transparency between physical scientists, behavioral scientists    , media, the public, policy-makers, NGOs, Indigenous peoples, and other stakeholders is essential to ensure that an ethical governance framework is established (Dowell et al., 2020; Diamond et al., 2022; Data for Progress, 2023; Carbon180, 2022). There must be effective engagement and education to co-create research questions and iteratively communicate research findings and results, as well as risks and co-benefits.    Failure to do so jeopardizes the trust and sound decision making of communities and governments (The Arctic Institute, 2021), threatening our ability to critically and openly assess potential climate solution approaches through a scientific process. Ideally, an external governance framework would be    developed which is enforceable and legally binding; however, there is also value in internal governance frameworks which may, for example, be based around a code of conduct, advisory or review boards, or other non-binding structures.

**Table 1: Key research and development activities for a five-year timeline beginning in 2023. Funding is needed for all research and development activities, including those that are already underway.**

| Category | Research and development activity | Year in which work becomes a funding priority | Anticipated duration (variable depending on outcomes) | Academia | Non-profits | Government | Engineering consultant |
|---|---|---|---|---|---|---|---|
| **Key milestone question #1: Is enhancement of the chlorine oxidative sink of methane via the iron salt aerosol (ISA) mechanism effective and climate beneficial? At what scale?** | | | | | | | |
| Modeling studies | Atmospheric plume and high-resolution regional modeling to understand if ISA-driven chlorine radical production results in net methane loss using various ISA sizes and concentrations of iron, NO., and other chemical species. | 2023 | 3 years | ☑ | ☑ | ☑ | |

| Category | Description | Year | Duration | | | | |
|---|---|---|---|---|---|---|---|
| | Global atmospheric modeling to understand if ISA-driven chlorine radical production results in net cooling, considering radiative forcing from methane, ozone, and iron aerosols, the photoactivity of the iron, meteorological and atmospheric chemistry conditions, and ISA size. | 2023 | 4 years | ☑ | ☑ | ☑ | |
| | Paleo modeling to understand variability in the chlorine-based methane sink that may have been associated with past dust changes. | 2024 | 2 years | ☑ | ☑ | ☑ | |
| Laboratory studies | Chamber studies to understand the photoactive iron fraction, chlorine radical production efficiencies, methane oxidation efficiencies, and reactions with other species. | 2023 | 3 years | ☑ | ☑ | ☑ | |
| Field studies | Natural analogue studies to (a) better constrain the atmospheric radical sinks; (b) understand the spatial extent and magnitude of the natural analogue, MDSA; and (c) understand the sources and distribution of photoactive iron (from MDSA as well as combustion iron sources). | 2023 | 3 years | ☑ | ☑ | ☑ | |
| | *In-situ* ISA enhancement field studies (e.g. ship plume and/or ground based) at a controlled experiment site to measure iron speciation and changes in chlorine, methane, and other species. | 2025 | 2+ years | ☑ | ☑ | ☑ | |
| Engineering | Initial design study for ISA sprayer system. | 2024 | 1 year | ☑ | | | ☑ |
| **Key milestone question #2: What impacts could the ISA approach have on Earth systems and human systems, both positive and negative? Is there a cost-plausible deployment method?** | | | | | | | |
| Modeling studies | Modeling of ocean system impacts considering different release locations/timing, ISA sizes, and atmospheric conditions. | 2024 | 3+ years | ☑ | ☑ | ☑ | |

| | | | | | | | |
|---|---|---|---|---|---|---|---|
| | Modeling of ISA deposition impacts on different land and ice surfaces considering different release locations/timing, ISA sizes, and atmospheric conditions. | 2024 | 3+ years | ☑ | ☑ | ☑ | |
| | Modeling of atmospheric impacts including clouds, precipitation, and stratospheric ozone considering different release locations/timing, ISA sizes, and atmospheric conditions. | 2024 | 3+ years | ☑ | ☑ | ☑ | |
| | Lifecycle analysis to quantify ISA climate benefits, tradeoffs, and cost. | 2024 | 4+ years | ☑ | ☑ | ☑ | |
| Governance and social impacts | Study of potential human health impacts considering different release locations/timing, ISA sizes, and atmospheric conditions. | 2025 | 3 years | ☑ | ☑ | ☑ | |
| Engineering | Advanced design study for ISA sprayer system. | 2026 | 1 year | ☑ | | | ☑ |
| | Design study of deployment modalities and implementation scenarios. | 2027 | 2 years | ☑ | | | ☑ |
| **Key milestone #3: What is needed to advance a structure of ethical governance and social license for utilizing atmospheric intervention to reduce atmospheric methane concentrations?** | | | | | | | |
| Governance and social impacts | Engage with stakeholders regarding the research findings and results, risks, and potential. | 2023 | 4+ years | ☑ | ☑ | ☑ | |
| | Develop a collaborative governance framework to monitor and report on environmental and social impacts. | 2024 | 4+ years | | ☑ | ☑ | |
| | Stakeholder engagement for identification of potential deployment locations and strategies. | 2026 | 3 years | | | ☑ | ☑ |

314

315

316

### 3.3 Priorities and timeline

It may be advantageous to pursue multiple research questions in parallel because the output from one research question may inform the inputs for other research questions. As such, activities can be sequenced using a prioritized timeline (Table 1), where later research activities and action areas often have multiple dependencies on earlier activities. For example, the engineering design study is suggested to begin in year 4 (2027) because it can start prior to having complete Earth system modeling results, human health study outcomes, or conclusions from ISA enhancement field studies. However, the engineering design study cannot advance to its later stages until earlier activities have been thoroughly addressed. This expedited schedule advances possible timelines to avoid delaying potential deployment-readiness, but is not meant to accelerate the timeline beyond appropriate caution and due diligence. Setbacks in addressing early research questions and action areas will likely result in timeline delays.

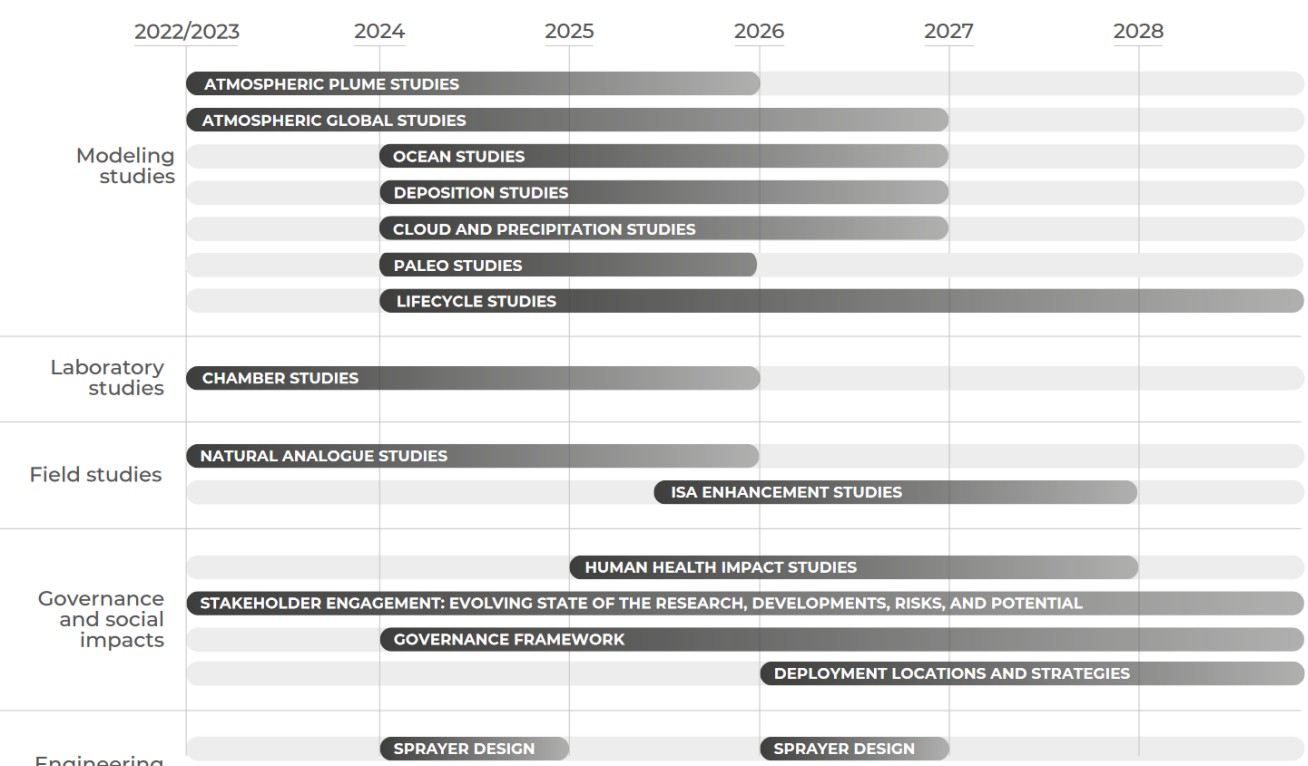

**Figure 1: Roadmap of ISA research and development needs. Duration of research and development timelines may be longer or shorter than depicted and/or exceed the five-year horizon.**

## 4 Conclusion

The activities outlined in this roadmap require coordinated efforts across multiple government agencies to financially support research and development, to ensure robust assessment and governance processes, and to foster international engagement. This work is valuable for multiple reasons:

1. We need to understand if ISA is a feasible, scalable, and safe methane removal method, or if it is nonviable and further research should be deprioritized.
2. Though this roadmap is ISA-specific, the research and development needs identified here contribute to fundamental understanding of processes and mechanisms that are broadly applicable to exploration of other methane removal approaches.
3. The research outlined in this roadmap will contribute to constraining the global methane budget and oxidative character of the atmosphere, which will improve our understanding of atmospheric chemistry, Earth system dynamics, and air quality.

Addressing the climate crisis requires a diverse portfolio of climate solutions. It is essential that atmospheric methane removal approaches are researched      in addition to, not replacing, crucial anthropogenic greenhouse gas emission reductions and carbon dioxide removal. All atmospheric methane removal approaches are at a very early stage (Jackson et al., 2021; Ming et al., 2022; Spark, 2023); all require further research and none are ready for deployment. We hope that this ISA roadmap, and other atmospheric methane removal roadmaps that follow, will help accelerate, prioritize, and parallelize research that is essential to understanding which climate solutions to pursue.

**Author contribution**

KG and SA wrote the manuscript draft. TJ, PH, NM, DM, MJ, MvH, YX, AS-L, TR, CB, ER, and DM contributed to, reviewed, and edited the manuscript.

**Competing Interests**

Spark Climate Solutions has in the past and anticipates in the future making research grants in alignment with this roadmap.

**Acknowledgements**

We thank two anonymous reviewers whose thoughtful and constructive comments improved the paper. We also thank Romany Webb, Sabine Fuss, Jean-François Lamarque, Paige Brocidiacono, Eric Davidson, Rob Jackson, and Celina Scott-Buechler for helpful discussions during preparation of the manuscript.

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
