# Peer review of "Opinion: A research roadmap for exploring atmospheric methane"

_EGUsphere, 2023_

## Author Comment (AC1)

**Response to reviewers**

Gorham et al., Opinion: Exploring potential atmospheric methane removal approaches: an example research roadmap for chlorine radical enhancement

**Reviewer 1**

**This 'opinion' paper lays out a road map for exploring the feasibility of injecting iron salt aerosol (ISA) to increase Cl atom concentrations in the lower troposphere and thus increase the removal rate of methane. There has been a lot of murmur in the community recently about proposals to accelerate methane removal by increasing tropospheric chlorine, and it is nice to see this paper provide a review of where things stand and outline a research program to study the possibility further. I find the paper to be well informed and generally well written though with some platitudes and repetitions. The abstract is a fair representation of the paper. I recommend publication but have several comments for the authors' consideration.**

We thank the anonymous reviewer for their thorough, constructive comments. Responses to each are below.

1. **This is effectively a white paper for a research program. The first authors are from Spark Climate Solutions, which funds research, so the paper probably describes a current or planned Spark research program rather than being a grass-roots opinion piece. It would be good to have transparency by stating upfront the role of Spark and its plans toward this research program.**

   The reviewer is correct that this is effectively a whitepaper for a research program. Spark Climate Solutions intends to fund aspects of this research.

   Line 369-370: Added "Competing Interests: Spark Climate Solutions has in the past and anticipates in the future making research grants in alignment with this roadmap."

2. **In the same vein, I don't see the point of the word 'example' in the Title.**

   Line 1-4: We have removed the word 'example' from the title. The title has been changed from "Opinion: Exploring potential atmospheric methane removal approaches: an example research roadmap for chlorine radical enhancement" to "Opinion: A research roadmap for exploring atmospheric methane removal via iron salt aerosols"

   Line 34: We have removed the word "example".

3. **Figure 1 does not add anything to Table 1 and could be deleted.**

   While we agree that Figure 1 and Table 1 have overlapping content, we believe that it is important to retain Figure 1 as a visual representation, recognizing that this format may be more accessible for some readers.

4. **Please scour the paper for opportunities to make the text more concise.**

   We have made a number of edits and deletions to make the text more concise:

Line 57-60: Remove this text "One estimate is that natural emissions could increase by 20% (~50 Tg CH4/yr) to 150% (~330 Tg CH4/yr) by 2100, becoming larger than anthropogenic emissions in most modeled scenarios (Kleinen et al. 2021)." and replaced with "On our current trajectory, natural emissions are estimated to increase by ~30-200 Tg $CH_4$/yr by 2100 (Zhang et al., 2023; Kleinen et al., 2021)."

Line 113: Replaced "The semi-arid regions of North Africa are the…" with "North African semi-arid regions are the…"

Line 114-115: Replaced "Using air samples collected during North African dust events from 1996-2000 in Barbados (Mak et al., 2003), a model…" with "Using air samples collected in Barbados during North African dust events (Mak et al., 2003), a model…".

Line 269-270: Replaced: "...marine life implications which could have far reaching ecosystem impacts, as well as economic (e.g. fisheries) and cultural (e.g. indigenous practices) impacts for coastal communities." with "...marine life implications resulting in economic, biodiversity, and cultural impacts for coastal communities."

5. **Although I enjoyed reading the paper and getting up to speed on ISA ideas, I'm still not sold at all on the idea that we need accelerated removal of methane. It's a very different matter for CO2 which has a much longer lifetime – even if we go to carbon zero we will still need to remove CO2 from the atmosphere to achieve near-term climate stabilization. There is no such need for methane, except perhaps if feedback from natural sources leads to a situation where increasing methane is outside our control. That's a reasonable position and the current intro hints at it but I would like to see a clearer statement of why the authors think that accelerating the methane sink is an 'essential' part of the climate solution as stated in lines 341-342.**

We added clarification to the text.

Line 83-84: Added "While not a replacement for much needed anthropogenic emission reductions, enhancing atmospheric oxidative sinks may be an important negative emission approach considering projected ongoing elevated natural methane emissions."

The use of "essential" refers to the need to consider methane removal in addition to greenhouse gas emission reductions (i.e., not in place of). It is not a statement about methane removal itself being essential. We changed the word "considered" to "researched" to more clearly make this point.

Line 361: Changed "considered" to "researched".

6. **Line 46: 'Methane (CH4) has contributed 0.5 °C to the current 1.1 °C temperature increase…' is misleading because of the offsetting effect of aerosols. It would be fairer to say that methane has contributed one third of the greenhouse warming since preindustrial time.**

Line 47-48: We have updated the text to avoid this potential for misleading. It now reads: "Methane (CH4) emissions have contributed roughly 0.5C to global warming relative to preindustrial times, second only to carbon dioxide (CO2)." We believe that the absolute number (0.5C) is worth keeping in the text.

7. **Line 49: Methane is now triple preindustrial levels.**

Preindustrial methane has been estimated to be 722 ppb and we are currently at ~1920 ppb. This is a factor of 2.66 so we don't feel comfortable saying "triple."

Line 50: Added the word "over" before "2.5".

8.  **Line 59: also cite Qu, Z., D. J. Jacob, Y. Zhang, L. Shen, D. J. Varon, X. Lu, T. Scarpelli, A. Bloom, J. Worden, and R. J. Parker , Attribution of the 2020 surge in atmospheric methane by inverse analysis of GOSAT observations, Environ. Res. Lett., 17, 094003, 2022.**

Line 61: Added "Qu et al, 2022".

Line 573-575: Added "Qu, Z., Jacob, D.J., Zhang, Y., Shen, L., Varon, D.J., Lu, X., Scarpelli, T., Bloom, A., Worden, J., and Parker, R.J.: Attribution of the 2020 surge in atmospheric methane by inverse analysis of GOSAT observations, Envion. Res. Lett., 17, https://doi.org/10.1088/1748-9326/ac8754, 2022."

9.  **Line 64: the 11 Tg a-1 loss is for tropospheric chlorine, which is of course the focus of this paper, but is important to clarify because the loss from stratospheric chlorine is of comparable magnitude.**

We agree. This has now been clarified in the text.

Line 65-68: Added four occurrences of "tropospheric".

10. **Lines 200-201: Reaction of Cl with ozone conserves chlorine radicals (produces ClO). ClO will go back to Cl very quickly and so it doesn't represent an effective sink for Cl – the cited Wang et al. 2019 paper certainly does not argue that it would be. A more effective sink (from the Wang paper) would be deposition of HCl to the oceans.**

We have added this clarification that the deposition of HCl is the main effective sink to consider.

Line 214: Removed "e.g. to reactions with ozone instead".

Line 214-215: Addd " One potential mechanism by which the cycling efficiency could be reduced is if hydrochloric acid is produced and deposited into the ocean (Wang et al, 2019)."

Line 215-217: Removed "After reacting with ozone there is a much longer chain of reactions before hydrochloric acid is created, thus reducing the cycling efficiency if the hydrochloric acid is deposited (Wang et al., 2019)."

11. **Line 260: another negative impact would be for Cl to drift into urban areas and produce ClNO2 to stimulate urban ozone formation. Wang, X., D.J. Jacob, X. Fu, T. Wang, M. Le Breton, M. Hallquist, Z. Liu, E.E. McDuffie, and H. Liao, Effects of anthropogenic chlorine on PM2.5 and ozone air quality in China, Sci. Technol., 54, 9908-9916, 2020.**

We have added this potential negative impact and cited the Wang et al. paper.

Line 270-271: Added "Furthermore, if chlorine drifts into urban areas it could stimulate ozone formation and cause negative human health impacts (Wang et al., 2020)."

Line 637-639: Added "Wang, X., Jacob, D.J., Fu, X., Wang, T., Le Breton, M., Hallquist, M., Liu, Z., McDuffie, E.E., and Liao, H.: Effects of Anthropogenic Chlorine on PM2.5 and Ozone Air Quality in China, Environ. Sci. Technol., 54, 9908-9916, https://doi.org/10.1021/acs.est.0c02296, 2020."

12. **Lines 272-273: 'reduced hydroxyl radical production may increase lifetimes of volatile organic compounds that are detrimental to human health'. In principle yes, but I am hard pressed to find an example. Carcinogenic VOCs tend to have short lifetimes so would be oxidized within urban areas where this ISA chemistry would only have faint effect.**

**Benzene, maybe? But for such a long-lived VOC the sink in urban air is effectively ventilation, not reaction with OH.**

As far as we know, these effects have not been rigorously studied. We believe that it is important to mention the potential air quality impacts. We have revised the statement to clarify.

Line 229-290: Changed from "However, the reduced hydroxyl radical production may increase lifetimes of volatile organic compounds that are detrimental to human health." to "However, the reduced hydroxyl radical production may increase lifetimes of atmospheric trace species that may be detrimental to human health."

13. **Section 3.2.3: seems to me that this section is not really saying anything beyond vague generalities. Make it more substantive?**

We have consolidated this section to be more substantive and concise.

Line 306-311: Removed "Policy and social science aspects of this roadmap must occur in parallel with the research and technical activities and should be iterative, where outcomes from policy and social science efforts inform development of research and technical direction, and vice versa. A misaligned or siloed approach may result in outcomes where physical science results are not answering the social science questions to understand if there could be a safe, cost-plausible deployment method and pathway to ethical governance, which may stall or halt progress. Conversely, detrimental outcomes could also be realized if research and technical aspects were delayed."

Line 313-314: Revised to "It is imperative that governance and social impacts be considered in parallel and iteratively with the development of any atmospheric methane removal approach."

Line 325-326: Removed "If ISA remains promising, potential implementation scenarios could be considered. Some methane removal solutions may be deployed immediately, whereas others may be higher risk and may only be suitable for emergency deployment."

**Reviewer 2**

**This opinion article presents a case for considering broad research goals and timelines coordinated across disciplines for the topic of atmospheric methane removal via Iron Salt Aerosols (ISA). The natural analogue of ISA, mineral dust sea salt aerosol (MDSA) interaction, is proposed to produce chlorine radicals that can react with methane based on lab studies and isotope measurements at ground-based stations. The article includes discussion of the effectiveness, side effects, scalability, and ethical governance needs for ISA to be considered a viable method for methane removal. By outlining science, engineering, and social science priorities, this manuscript does indeed propose a roadmap for this potential technology.**

**I find the article to be compelling and of thorough scope, well-researched and appropriately cautious while aiming to push science toward technological solutions to curb climate change. The article should find interest from the audience reached by ACP, and could likely stoke discussion and research on the outlined priorities as intended. Figures and text are presented clearly and with sufficient detail. I include below one suggestion for a non-critical addition as well as a few points for clarification; otherwise, I consider this opinion piece to be a solid candidate for publication in ACP.**

We thank the anonymous reviewer for their constructive comments, addressed below.

**Major comment:**

I would be interested to understand the role of roadmapping in atmospheric science, historically and/or presently. The authors implicitly make the case that this practice will accelerate society's ability to translate science into technological solutions for climate change. Have other organizations or entities taken such a targeted approach to strategizing about a research portfolio in support of a technological development in a geophysics field? The approach certainly makes sense to me; I just wonder if there are examples that point to increased success, accelerated timelines, etc. associated with this practice. I don't consider this addition to be critical, just an opportunity, perhaps, to bolster the justification for this methodology.

We agree with the reviewer that a better understanding of roadmapping in atmospheric science (and particularly potential climate interventions) would be an additional contribution to the literature. At present, the existing roadmaps in this space are quite recent. We have cited a number of them (Wanser, Maesano, Aschwanden, Ocean Visions) in Section 3.1, all of which were published in 2021 or later. It is too early to make definitive statements about the utility of roadmaps, we believe. We have added some clarifying details to the roadmaps we cite as examples describing their topics in Section 3.1.

Line 174-178: Revised to now read, "..., particularly in instances that benefit from integrated, interdisciplinary research. Recent examples include geochemical carbon dioxide removal (Masano et al., 2022), ocean-based carbon dioxide removal (Ocean Visions, 2023), ice sheet contributions to sea level rise (Aschwanden et al., 2021), and solar radiation management ( (e.g. Wanser et al., 2022)."

**Minor comments:**

**L77: Could cite Holmes, 2018 for a thorough discussion of the methane feedback phenomenon: https://doi.org/10.1002/2017MS001196**

Line 80: Added "Holmes, 2018"

Line 449-450: Added "Holmes, C.D.: Methane Feedback on Atmospheric Chemistry: Methods, Models, and Mechanisms, J. Adv. Model Earth Sy., 10, 1087-1099, https://doi.org/10.1002/2017MS001196, 2018."

**L119: Perhaps it is worth naming the four ground-based stations in the North Atlantic, for interested readers.**

Line 125: We have added the station names, Barbados, Canary Islands, Cape Verde, and Brazil.

**L172: Some language in this section is repetitive, could be streamlined. For instance, I first latched onto these three assessment criteria: feasibility, scalability, and social license to operate. These don't exactly marry up to the questions/subsections below. Similarly, some of the sections below touch on topics from the other sections (e.g., ~L240, social license issues brought up in the "effectiveness" section).**

We're defining viable as "feasible", "scalable", and "with social license to operate". Feasibility is then made up of climate beneficial, safe, and cost-plausible. Milestone question 1 addresses climate beneficial and scalable, milestone question 2 addresses safe and cost-plausible, and milestone question 3 addresses social license to operate.

There is some overlap between the milestone question as the reviewer correctly points out. We've added a reference within Section 3.2 to clarify these connections.

Line 255: Added "(see Section 3.2.3)"

**L286: Could the authors clarify what is meant by this sentence, "To be cost-plausible…," for those who are not familiar with socioeconomic terms? Does this mean one must compare the cost of preventing emission of methane in the first place against the cost to remove it via ISA?**

We have added text to clarify.

Line 302-304: Changed "To be cost-plausible, these costs at scale must be lower than the social cost…" to "To be cost-plausible, the cost per ton of methane removed must have a viable path to becoming lower than the social cost…".

**Table 1: The use of "ongoing" in the "Anticipated duration" column is a bit confusing to me, suggests the same as "underway" used in the prior column. Does this just mean the task is of indeterminate length?**

We have removed all references to "underway" and "ongoing" and replaced them with quantitative values. We agree that this was unclear.

**I see now from Fig. 1 that the "ongoing" items in Table 1 are meant to continue throughout the duration of the timeline; perhaps "continuous" or some other wording would be better? Or can the use of "+" elsewhere in this column be adapted for this (i.e., wouldn't "5+ years" convey the same thing as "ongoing" in most of these cases)?**

We agree. See previous response.

**Additional revisions**

Line 143-244: Added "..., and lowering the global methane burden by 2,000 Tg would require the emission of an additional 1,000 Tg $Cl_2$ /yr."

Line 200: Removed "(van Herpen et al., 2023; Meidan et al, submitted)".

Line 200: Replaced "2%" with "1.8%"

Line 204-207: Added "Furthermore, the rate of chlorine production - and subsequent rate of methane oxidation - per mass of photoactive iron is estimated to result in the removal of 45 methane molecules per iron atom per day (van Herpen et al., 2023), but has many uncertainties including the time that iron remains in the atmosphere which may be impacted by large regional variability in deposition rates (Meidan et al., submitted)."

Line 207-209: Revised text to now read, "The efficiency, cost, safety (e.g. air quality), and net radiative forcing of ISA will depend on the percentage of iron that is photoactive, the rates of chlorine production and methane oxidation per mass of photoactive iron, and the lifetime of photoactive-iron based aerosol."

Line 315: Changed "scientists" to "physical scientists".

Line 315: Changed "behavioral scholars" to "behavioral scholars".

Line 321: Changed "is" to "would be".

Line 327-328: Moved title for Table 1 to above table.

Table 1: Under Key Milestone question #1, Field Studies, changed "*In-situ* ISA enhancement field studies (e.g. ship plume and/or ground based) at a controlled experiment site to measure changes in chlorine, methane, and other species." to "*In-situ* ISA enhancement field studies (e.g. ship plume and/or ground based) at a controlled experiment site to measure iron speciation and changes in chlorine, methane, and other species."

Line 366-368: Added author contribution.

Line 371-374: Added acknowledgements.

---

## Author Response (AR2)

**Response to editor**

Gorham et al., Opinion: Exploring potential atmospheric methane removal approaches: an example research roadmap for chlorine radical enhancement

*Dear Authors,*
*Thank you for answering the reviewer comments and improving the manuscript. The response is adequate and I am pleased to accept the revised manuscript as an Opinion article in ACP after the following change has been made. The motivation for the proposed work should be briefly reiterated in the conclusions section as to why it is of interest to investigate methane removal approaches rather than just focusing on reducing methane emissions that we can control (see Reviewer #1's comment 5). As long as anthropogenic methane sources dominate, targeted emission reductions can probably limit and even reduce atmospheric methane more easily and with less risk. However, if natural sources (e.g. permafrost thawing) become uncontrollable due to climate change, an artificial methane sink as discussed in the paper could become an option to stabilise methane levels. This point is worth mentioning again in the conclusions.*
*Best regards,*
*Andreas Hofzumahaus*
* * *
We thank the editor for their thoughtful comments, and for directing our attention to revisit Reviewer #1's comment 5. We have revised and expanded content in the conclusion section accordingly (see Author Track Changes).

Line 343-345: Revised "It is essential that atmospheric methane removal approaches are researched in addition to, not replacing, crucial anthropogenic greenhouse gas emission reductions and carbon dioxide removal" to "Atmospheric methane removal approaches should only be researched in addition to, not replacing, crucial anthropogenic greenhouse gas emission reductions and carbon dioxide removal."

Line 345-348: Added "Atmospheric methane removal approaches could play a future role in overall climate change mitigation alongside aggressive anthropogenic emissions reductions, for example by dampening the impacts of anthropogenically-amplified natural methane emissions (e.g., from wetlands or permafrost thawing), particularly if they become uncontrollable due to climate change."